# Feasibility of a Physiatry Assessment Clinic to Address Physical Impairment in Head and Neck Cancer Patients Following Neck Resection and Free Flap Reconstruction

**DOI:** 10.3390/curroncol32100562

**Published:** 2025-10-07

**Authors:** Lauren C. Capozzi, Chad Wagoner, Julia T. Daun, Lisa Murphy, Steven C. Nakoneshny, George J. Francis, Joseph C. Dort, Khara Sauro, S. Nicole Culos-Reed

**Affiliations:** 1Cancer Rehabilitation, Supportive Care, BC Cancer, Kelowna, BC V1Y 5L3, Canada; 2Department of Kinesiology, Recreation, and Sport Studies, University of Tennessee, Knoxville, TN 37996, USA; cwagone2@utk.edu; 3Faculty of Kinesiology, University of Calgary, Calgary, AB T2N 1N4, Canada; jtdaun@ucalgary.ca (J.T.D.); nculosre@ucalgary.ca (S.N.C.-R.); 4Division of Physical Medicine & Rehabilitation, Cumming School of Medicine, University of Calgary, Calgary, AB T2N 1N4, Canada; lisa.murphy2@ucalgary.ca (L.M.);; 5Department of Surgery, Cumming School of Medicine, University of Calgary, Calgary, AB T2N 1N4, Canada; scnakone@ucalgary.ca (S.C.N.); jdort@ucalgary.ca (J.C.D.); kmsauro@ucalgary.ca (K.S.); 6Community Health Sciences, Cumming School of Medicine, University of Calgary, Calgary, AB T2N 1N4, Canada; 7Department of Oncology, Cumming School of Medicine, University of Calgary, Calgary, AB T2N 1N4, Canada; 8Supportive Care, Psychosocial and Rehabilitation Oncology, Cancer Care Alberta, Alberta Health Services, Calgary, AB T2N 1N4, Canada

**Keywords:** head and neck cancer, free flap reconstruction, neck resection, cancer physiatry, cancer rehabilitation, impairment-driven screening

## Abstract

**Simple Summary:**

Individuals with head and neck cancer are living longer than ever before, yet due to intensive treatments including surgery, many live with long-term side effects impacting their function and quality of life. A growing body of evidence supports the role of early rehabilitation in improving patient outcomes; however, the optimal timing and implementation of rehabilitation services have not been established. This study assessed the feasibility of a physiatry assessment clinic approximately two months postoperatively. The physiatry assessment clinic was found to be challenging for people to attend, and did not necessarily target those with higher needs. These findings reinforce the importance of established rehabilitation screening programs that identify those in need of and facilitate triage to appropriate rehabilitation services.

**Abstract:**

Individuals with head and neck cancers are living longer than ever before, yet many live with the long-term effects of their cancer and treatment. The purpose of this study was to assess the feasibility of a physiatry assessment clinic (PAC) following neck resection and free flap reconstruction, during which physical function was assessed. Methods: Adult patients participating in a larger prehabilitation study were included. Attendance and the ability to complete the physical function assessment were examined. Exploratory analyses were completed to describe physical function, fitness, shoulder, and neck function among PAC attenders. To further understand PAC feasibility, patient-reported outcomes among PAC attenders and non-attenders were examined over 12 months (QuickDASH, NDII, EAT-10). Results: A total of 36 eligible participants (78.2%) from the larger prehabilitation study were approached to participate in the PAC, and 19 of the 36 attended (52.8%). Participants attended on average 8.6 ± 3.6 weeks post surgery, and 100% were able to complete the functional measures. Exploratory data suggest that those who did not attend (17 of 36 approached) had more advanced disease compared to those who attended (*p* < 0.05). Patient-reported outcomes suggested better shoulder function and swallow function at 6 months among those who attended the clinic versus those who did not. Conclusions: While recruitment to the PAC and assessment completion demonstrated feasibility, attendance posed challenges for patients. These findings highlight the need for innovative approaches to screening patients and tailoring rehabilitation services based on physical impairment.

## 1. Introduction

Head and neck cancer (HNC) is the seventh most common type of cancer worldwide [1]. Due to improved screening, detection, and treatment, as well as higher rates of HPV-associated HNC, survival rates have improved significantly over the last 45 years, from 54.6% in 1975 to 78% in 2018 [2]. Treatment for HNC often includes major surgery involving tumor removal, lymph node dissection, and free flap reconstruction. Although often lifesaving, surgical management often removes soft tissues and bone, which can have detrimental effects on neck and shoulder function, speaking, chewing, and swallowing [3,4]. This type of surgery is complex, and many patients experience postoperative morbidity and complications, which negatively affects function and quality of life (QOL) [4,5,6,7,8,9]. Patients with neck dissection often have reduced cervical spine and shoulder mobility, as well as deficits in neck and shoulder strength [3]. These impairments cause many patients to experience reductions in their self-confidence, dignity, quality of life, and ability to return to work [4].

Growing evidence shows that early rehabilitation interventions can improve patient outcomes following a cancer diagnosis [10]. Understanding the timing and types of impairments experienced by patients undergoing surgery for HNC can inform tailored interventions. Previous work in patients with other tumors shows that multidisciplinary physiatry (physical medicine and rehabilitation) assessment clinics can improve the identification of physical impairment and enable targeted rehabilitation interventions. However, there is little evidence exploring the feasibility and role of a physiatry clinic for patients with HNC following neck resection with free flap reconstruction [11].

Therefore, the purpose of this study was to (1) assess the feasibility of a postoperative physiatry assessment clinic (PAC) for patients with HNC, including the ability to attend the assessment approximately 4–6 weeks following surgery and complete the assessment measures, and (2) describe functional impairment among patients with HNC following surgery to inform future fully powered trials. We hypothesized that a postoperative PAC would be feasible, targeting patients with functional impairment at an appropriate timepoint.

## 2. Materials and Methods

### 2.1. Study Design and Setting

This study, examining the feasibility of a PAC and postoperative neck and shoulder function in HNC, was nested within a larger parent study. The parent study was a hybrid implementation/effectiveness trial, examining a multiphasic exercise prehabilitation intervention that was initiated before surgery and continued in hospital, through to discharge, and one year postoperatively [12].

### 2.2. Ethical Approval

Study approval was obtained from the Health Research Ethics Board of Alberta—Cancer Committee (HREBA-CC)—HREBA.CC-20-0013 [12]. The clinical team (the surgeon or clinical nurse) briefly introduced this study to eligible patients in the initial clinic appointment and gained consent to contact. The study team then contacted interested patients to review study eligibility, details of this study, and obtain informed consent to participate. Participants were then sent a link to Research Electronic Data Capture (REDCap), providing access to a secure database for documentation of written informed consent [13]. All patients who consented to the parent study also consented to a referral to the PAC. The PAC accepted referrals from February 2022 to September 2023, and therefore all patients involved in the larger parent study during this period were provided with an opportunity to participate in the PAC. Patients not involved in the parent study were not eligible for the PAC.

### 2.3. Participants and Sample Size

All newly diagnosed HNC patients over the age of 18 undergoing surgical resection with free flap reconstruction were eligible for the larger parent study [12]. Inclusion criteria included the following: 18 years of age or older, histologically verified primary head and neck carcinoma, scheduled to undergo oncologic resection with free flap reconstruction, with approval received from a clinical exercise physiologist and/or clinician, and an ability to provide written informed consent and understand study information in English. Individuals were excluded if they had concurrent neurologic or musculoskeletal comorbidity inhibiting exercise; were diagnosed with a psychotic, addictive, or major cognitive disorder; had severe coronary artery disease (Canadian Cardiovascular Society Class III or greater); had significant congestive heart failure (New York Heart Association Class III or greater); or had an active infection. At the time of initial recruitment, participants were also asked if they would like to attend a postoperative PAC (4–6 weeks postoperatively). Reasons for not attending were tracked and characteristics of both attending and non-attending groups were recorded to assess for feasibility of the PAC. Of those who attended, assessment completion was tracked to assess feasibility.

### 2.4. Physiatry Assessment Clinic (PAC)

The PAC was led by a physical medicine and rehabilitation (physiatry) resident physician and a clinical exercise physiologist. Participants were booked for 45 min appointments, during which their medical and functional histories were reviewed, a central and peripheral neurological examination was conducted, a shoulder and neck examination was performed, and the Short Physical Performance Battery protocol (SPPB) was performed [13]. Participants also completed patient-reported outcomes prior to attending the PAC (described below).

### 2.5. Variables and Data Management

Feasibility of the PAC was the primary outcome, assessed by comparing characteristics between those who attended versus those who did not attend. Exploratory analyses were completed to describe physical function, fitness, and shoulder, neck, and swallowing function between those who did and did not attend the PAC to understand whether the PAC recruited the participants with impairment at a time of need and if it was feasible for those with higher needs to attend the PAC.

#### 2.5.1. Measures of Feasibility

Measures of feasibility included the proportion of eligible participants who attended the PAC, and was set at a minimum of 60%. This value was selected based on previous work in this area [11,14]. Understanding characteristics among those who did and did not attend the PAC were collected to identify potential barriers to attending. In addition, the percent of PAC assessments completed was recorded. Any adverse events during the PAC were also evaluated.

#### 2.5.2. Demographic and Clinical Characteristics

Demographic and medical variables were collected via chart review and a baseline questionnaire. Patients reported age, sex, gender, education, family income, employment status, racial background, smoking, and alcohol status. Clinical characteristics including time since surgery and details of diagnosis (cancer site, disease pathology, staging, surgical side, and adjuvant treatment details) were obtained by chart review prior to each patient’s PAC appointment.

#### 2.5.3. Physical Function

The PAC evaluation included a complete medical history, including vital signs (heart rate and blood pressure), anthropometrics (height, weight, and body mass index), physical function (short physical performance battery), and assessment of neck and shoulder range of motion and strength. The neck assessment included range of motion testing using a goniometer for neck flexion, extension, lateral rotation, and lateral flexion. Neck flexion and extension strength were recorded. The shoulder assessment included inspection for scapular winging, and range of motion using a goniometer for shoulder flexion, abduction, and internal and external rotation (with the arm abducted to 90 degrees). Shoulder strength with upper extremity myotomes (C5 to T1), and spinal accessory nerve-innervated muscle (sternocleidomastoid and trapezius) strength was tested. All muscle strength was graded as per the Medical Research Council (MRC) scale for muscle strength [15].

#### 2.5.4. Short Physical Performance Battery (SPPB)

The SPPB consists of a group of three physical examination tests evaluating gait speed, chair stand speed, and balance testing [13]. It is a validated tool, used to predict risk for mortality, nursing home admission, and disability, and has been previously used in cancer rehabilitation assessment clinics [11,13]. It is scored from 0 (worst performance) to 12 (best performance).

#### 2.5.5. Patient-Reported Outcomes (PROs)

##### Shoulder and Arm Function Outcome Measure

Shoulder function was assessed using the QuickDASH. The QuickDASH is an 11-item patient-reported outcome (PRO) measure of functional limitations and symptoms (pain, pins and needles) in the upper limb [16]. The scores range from 0 to 100, with 100 representing total impairment. The QuickDASH has demonstrated both internal consistency and test/retest reliability in women with breast cancer and construct validity in an orthopedic population [17].

##### Shoulder and Neck Function Outcome Measure

Neck function was assessed using the Neck Dissection Impairment Index (NDII), a 10-item region-specific PRO measure of quality of life (QOL) validated in patients who have undergone neck dissection [18]. A summary score from 0 (worst QOL) to 100 (best QOL) is calculated from reported shoulder and neck-related pain and stiffness, problems with self-care activities, and participation in social activities, recreational activities, or work.

##### Swallowing Function

Swallowing symptomatology was assessed using the EAT-10, a 10-item symptom-specific outcome instrument for dysphagia that has displayed excellent internal consistency, test/retest reproducibility, and criterion-based validity [19].

### 2.6. Statistical Analysis

All data were analyzed using R-Studio Version 4.3.0 (Boston, MA, USA). Descriptive statistics (frequencies with proportions, means with standard deviations) were used to describe patient demographics and clinical variables, and independent *t*-tests (continuous) and chi-square tests (categorical) were used to determine between-group differences across all outcomes among the attended versus not attended groups. Differences within groups for PROs across timepoints (baseline, 6 months, and 12 months) were examined using repeated measures ANOVA, with pairwise comparisons as appropriate. Given the clinical impact of the surgical side (i.e., left or right) on physical outcomes, the clinical assessment results were examined according to the surgical versus non-surgical side. For example, right shoulder abduction strength was categorized as ‘surgical’ if the surgical side was also on the right. Of those patients who had bilateral neck dissections, data were considered to be independent, and data from both the left and right sides were coded as ‘surgical’. The inherent risk of considering these data to be independent was offset by the clinical impact of the affected versus unaffected variable categorization.

## 3. Results

### 3.1. Demographic and Clinical Characteristics

See Table 1 for participant demographics and Table 2 for participant clinical characteristics among those who attended and did not attend the PAC. During the PAC recruitment period, a total of 36 individuals in the parent study were eligible and were approached. Of the 36, 19 participants attended the PAC. The mean age of participants who attended the PAC was 61.0 ± 14.2 years and the majority of participants were male (n = 15, 78.9%). There were no differences in baseline demographic characteristics between those who attended the PAC and those who did not attend.

Of those who attended the PAC, the average time since surgery was 8.6 ± 3.6 weeks, and the most commonly diagnosed cancer was squamous cell carcinoma (n = 15, 78.9%) of the oral cavity (n = 14, 73.7%), with the majority being stage IVA or B (n = 7, 36.8%). Significant differences in cancer stage were observed between those who attended versus those who did not attend (*p* < 0.05), with the group who attended having a lower stage of disease.

### 3.2. Feasibility

See Figure 1 for the flow of patients in this study. A total of 36 participants were approached to participate in the PAC and a total of 19 attended the clinic (52.8%). Of those who attended, 100% completed the assessment. There were no adverse events reported during the PAC. Reasons for not attending are included in Figure 1.

### 3.3. Shoulder and Neck Function

The clinical assessment results of those who attended the PAC are outlined in Table 3 and Table 4. Of those who attended, the mean resting heart rate and blood pressure fell within normal ranges. The average BMI was 26.5 + 3.9 kg/m^2^. The average SPPB score was 11.0 ± 1.6/12. Table 4 outlines shoulder and neck function. As described in Section 2, instead of describing left versus right shoulder values, the results are described by surgical side to assess possible differences in function. There were no differences (*p* > 0.05) between the surgical and non-surgical sides when evaluating neck and shoulder strength, range of motion, and scapular winging.

### 3.4. Shoulder and Neck Function Across Timepoints and Between Groups

There were no significant differences in QuickDASH scores over time among the attended and not attended groups (Table 5). At the 6-month timepoint, however, there was a significant difference, with those who attended scoring lower on the QuickDASH (i.e., higher function) compared to those who did not attend, 14.9 ± 3.9 versus 19.2 ± 4.5 (*p* < 0.05). There were no differences across time or between groups on the neck function tests. There was an improvement in PROs reported on the EAT-10 from 6 weeks to 6 months postoperatively, with the attended group having lower symptom scores compared to those who did not attend.

## 4. Discussion

This study examined the feasibility of a PAC following neck resection with free flap reconstruction for individuals living with and beyond HNC. Although the a priori registration feasibility cut-off was met (i.e., 78.2%, which was greater than the pre-determined 50% cut-off), attendance among those who registered was 52.8%, which was below the a priori feasibility cut-off of 60%, suggesting the PAC in this patient population, approximately 2 months post surgery, was not feasible. While participants who attended the PAC completed all assessments, and there were no adverse events, exploratory data suggest that those who attended had relatively intact neck and shoulder function on physical examination and high functional scores on the SPPB. This contrasts with those who did not attend, who were found to have more advanced disease and lower shoulder function scores on the QuickDASH at 6 months post surgery. Taken within the context of the growing area of rehabilitation medicine, this work highlights the need for innovative approaches that engage patients undergoing neck resection with free flap reconstructive surgery who will most benefit from rehabilitation care (i.e., those with advanced disease with worse functional status) [20,21].

Rehabilitation care is critical following a cancer diagnosis and treatment, especially in patient populations with a high risk of function impairment, like those with HNC undergoing surgery [21,22,23,24]. As the evidence supporting rehabilitation grows, so do targeted resources, including cancer physiatry services, physiotherapy, occupational therapy, exercise programming, speech language pathology services, and many others [21,25]. Unfortunately, cancer care systems still struggle with methods to identify patients in need of rehabilitation, and the implementation of pathways to triage patients to the right rehabilitation services [22,26]. According to the current standard of care in our Centre in Calgary, Alberta, all patients who undergo head and neck resection with free flap reconstruction see an inpatient physiotherapist postoperatively and are referred to outpatient physiotherapy within 6 weeks of discharge. The proportion of patients who attend this outpatient physiotherapy appointment is unknown. It is also unknown if these appointments target those most in need of rehabilitation services. Previous work in the neuro-oncology population reinforced the benefit of a physiatry assessment and triage clinic in identifying physical and cognitive impairments among patients, thereby facilitating triage to appropriate services or providers [11]. It also highlighted the importance of impairment-driven care, and the value of screening to identify symptoms that can be addressed by rehabilitation professionals [11,27]. Impairment-driven care, which is a targeted approach to rehabilitative care that focuses on identifying functional impairments experienced by patients, is designed to ensure the right care is offered for the right person at the right time [22,28]. Impairment-driven care also has the opportunity to improve equity in cancer care, offering services based on need, instead of relying on patients to report functional impairments or busy oncology professionals to ask patients about functional concerns [29]. It also reserves care for those most in need, versus offering a blanket referral to all patients. This helps with resource management and improves personalized care for patients. Methods to facilitate impairment-driven care include screening for impairment and triage pathways to provide patients with access to the appropriate care based on their specific needs. There are a variety of rehabilitation screening tools and triage pathways that have been proposed in cancer care, including the Patient-Reported Outcome Measure Information System (PROMIS) Cancer Function Brief 3D Profile, the EXCEEDs algorithm, and the Cancer Rehabilitation and Exercise Screening Tool (CREST) [11,30,31]. Functional tools often used in the HNC population tend to be research-based questionnaires including the QuickDASH, EAT10, FACT-HN, and Neck Dissection Impairment Index (NDII) [16,17,19,32,33]. To the best of our knowledge, these tools have not been used in a clinical setting to screen for impairment and guide rehabilitative treatment. Validating one of these existing tools for screening purposes, or developing a new tool that targets specific impairments experienced by patients following surgery for HNC would allow for improved patient care and more targeted triage to necessary rehabilitation services. As evidenced from this study, not everyone requires rehabilitation following neck dissection, and therefore screening postoperatively for impairment may provide greater patient benefit, limit appointment burden on those who do not require rehabilitation, and ultimately be more cost-effective in identifying those who have needs and targeting these individuals specifically [34].

In addition to screening patients to identify those in greatest need of rehabilitative services, a second consideration is the timing of the services offered. Those who did not attend the PAC cited scheduling conflicts, busyness, and medical complications as key barriers to attending the clinic. Given the complexity of free flap reconstructive surgery, the recovery timelines for this surgery, and that these patients may also be undergoing adjuvant treatments for their cancer, the low attendance in the PAC may suggest that approximately 8–9 weeks following surgery is not a feasible timeframe for these patients. Interestingly, the PAC was originally scheduled to occur approximately 4–6 weeks postoperatively, with the goal of seeing patients prior to possible initiation of chemotherapy and/or radiotherapy. Compared to a prior study conducted in neuro-oncology using a similar multidisciplinary PAC, enrolment exceeded 60% and attendance was 94.7% [11]. Although this clinical population was also complex, with multiple comorbidities and functional impairments, they were significantly further from initial diagnosis (i.e., over 6 years on average), and the majority were not undergoing active treatment [11]. Given the paucity of evidence on PACs for patients undergoing neck resection with free flap reconstruction, it is challenging to determine what may be a more feasible timeframe for PAC appointments. However, a concern with delaying the PAC appointments includes missing impairments that could benefit from early rehabilitation intervention, like shoulder dysfunction secondary to spinal accessory nerve injury.

Future research should focus on the trajectory of impairments among patients with HNC undergoing neck resection with free flap reconstruction, to identify optimal timeframes for rehabilitation. Previous work has highlighted common impairments, including shoulder dysfunction, neck dysfunction, pain, dysphagia, trismus, fatigue, and deconditioning [6], but understanding the onset of these impairments and peak would allow for more targeted rehabilitation care. Future research should also include additional variables such as length of hospital stay and detailed description of donor site recovery, which were not collected in this study. Screening for impairment prior to surgical follow-up appointments may also help with tailoring the timeframe for referrals to rehabilitative services. Future work should also include refinement of short, impairment-oriented, patient-centered screening tools that can be completed as an outpatient electronically at set postoperative intervals (for example, at 6, 8, 12, and 24 weeks). Screening tools can use a Likert scale, and if high impairment is reported or trends with increased impairment are identified, patients can be automatically referred to a PAC. Here, a medical and physical assessment can be conducted (similar to that described in this study), which can facilitate referrals to providers (i.e., PT, OT, SLP, or continued physiatry care). Studies are underway to refine and validate a screening tool for impairment which may help to identify patients in need of a PAC appointment.

### Limitations

While this study is the first to test the feasibility of a PAC to identify impairments among HNC patients undergoing neck resection with free flap reconstruction who may benefit from rehabilitation interventions, there are limitations which should be considered. First, recruitment was limited to those who consented to participate in the larger prehabilitation study, which may have resulted in a homogeneous group of participants [12]. This may reduce the generalizability of these results beyond this context. Similarly, because cancer physiatry is a growing field, a PAC may not be offered in all settings. Additionally, this was a feasibility trial and was not powered to make between-group comparisons, and therefore differences between those who attended and did not attend the PAC should be interpreted with caution.

## 5. Conclusions

This study identified challenges with the feasibility of a PAC for HNC patients undergoing neck resection with free flap reconstruction. Specifically, attendance was low and it is possible that those with greater physical impairments did not attend. Patients who were unable, or chose not, to attend had higher rates of patient-reported impairment in the year following surgery, and also had a higher stage of disease. The findings of this study suggest that innovative approaches to screening patients for physical impairment and rehabilitation needs, and timely delivery of these services are needed among patients undergoing neck resection and free flap reconstruction to better identify those who would most benefit from a PAC and rehabilitation services [11,30,31,35,36,37].

## Figures and Tables

**Figure 1 curroncol-32-00562-f001:**
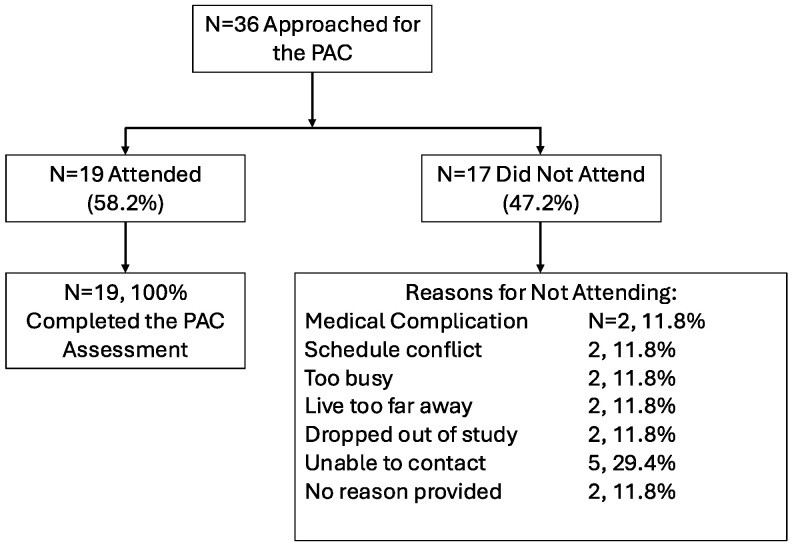
Physiatry clinic attendance.

**Table 1 curroncol-32-00562-t001:** Participant demographics (n = 36).

Demographic Variable	Total (n = 36)	Attended (n = 19, %)	Not Attended (n = 17, %)
Sex Male Female	26 10	15, 78.9% 4, 21.0%	11, 64.7% 6, 35.3%
Self-Identified Gender Man Woman	26 10	15, 78.9% 4, 21.0%	11, 64.7% 6, 35.3%
Age: Mean ± SD, *years*	62.5 ± 11.8	61.0 ± 14.2	64.5 ± 8.7
Highest Level of Education Some Secondary Education Completed High School Some University/College, No Degree Completed University/College Completed Graduate School Prefer Not to Answer	1 6 12 11 5 1	0 2, 10.5% 7, 36.8% 6, 31.6% 3, 15.8% 1, 5.26%	1, 5.88% 4, 21.1% 5, 29.4% 5, 29.4% 2, 10.5%0 0
Annual Family Income, *CAD* <CAD 20,000 CAD 20,000-39,999 CAD 40,000-59,999 CAD 60,000-79,999 CAD 80,000-99,999 >CAD 100,000 Prefer not to answer	2 3 6 5 3 11 6	1, 5.26% 2, 10.5% 3, 15.8% 4, 21.0% 1, 5.26% 5, 26.3% 3, 15.8%	1, 5.88% 1, 5.88% 3, 17.6% 1, 5.88% 2, 10.5% 6, 35.3% 3, 17.6%
Employment Status Unable to Work Due to Cancer Retired Part Time Homemaker Full Time	6 18 2 2 8	3, 15.8% 9, 47.4% 1, 5.26% 1, 5.26% 5, 26.3%	3, 17.6% 9, 52.9% 1, 5.88% 1, 5.88% 3, 17.6%
Distance From Clinic Distance in KM Distance in Minutes Travelled		44.9 + 59.5 37.3 + 34.6	90.9 + 113.2 64.8 + 67.9
Self-Described Racial Background White South Asian East Asian	32 3 1	17, 89.5% 1, 5.26% 1, 5.26%	15, 88.2% 2, 10.5% 0
Smoking Status Never Smoked Ex-Smoker Current Smoker Not Stated	14 13 7 2	6, 31.6% 8, 42.1% 4, 21.0% 1, 5.26%	8, 47.1% 5, 29.4% 3, 17.6% 1, 5.88%
Alcohol Use Never Consumed Alcohol Light Drinker Moderate Drinker Heavy Drinker Not Stated	8 10 2 10 6	3, 15.8% 5, 26.3% 2, 10.5% 6, 31.6% 3, 15.8%	5, 29.4% 5, 29.4% 0 4, 21.1% 3, 17.6%

**Table 2 curroncol-32-00562-t002:** Clinical characteristics (n = 36).

Clinical Characteristic	Total (n = 36)	Attended (n = 19, %)	Not Attended (n = 17, %)
Time Since Surgery: Mean ± SD, *weeks*		8.6 ± 3.6	
Primary Site of Head and Neck Tumor Oral Cavity Oropharynx Paranasal Sinus Skin Salivary Gland Larynx	28 2 1 3 1 1	14, 73.7% 2, 10.5% 1, 5.3% 1, 5.3% 1, 5.3% 0	14, 73.7% 0 0 2, 10.5% 0 1, 5.9%
Histology Squamous Cell Carcinoma Sarcoma Mucoepidermoid Carcinoma Adenoid Cystic Carcinoma Basal Cell Carcinoma Benign Not stated	28 2 2 1 1 1 1	15, 78.9% 2, 10.5% 1, 5.3% 1, 5.3% 0 0 0	13, 76.4% 0 1, 5.9% 0 1, 5.9% 1, 5.9% 1, 5.9%
Stage ***** (Using the American Joint Committee on Cancer 8, AJCC8) 0 I II III IV A or B Not Stated	1 3 6 5 19 2	0 1, 5.3% 5, 26.3% 5, 26.3% 7, 36.8% 1, 5.3%	1, 5.9% 2, 10.5% 1, 5.9% 0 12, 70.6% 1, 5.9%
pT Classification T0 T1 T2 T3 T4a Not Stated	2 4 9 9 11 1	1, 5.3% 2, 10.5% 6, 31.6% 8, 42.1% 2, 10.5% 0	1, 5.9% 2, 10.5% 3, 17.6% 1, 5.9% 9, 52.9% 1, 5.9%
pN Classification N0 N1 N2b N3 Not Stated	16 4 2 13 1	10, 52.6% 3, 15.8% 1, 5.3% 5, 26.3% 0	6, 35.3% 1, 5.9% 1, 5.9% 8, 47.1% 1, 5.9%
Surgical Side Left Right Bilateral	8 10 17	6, 31.6% 7, 36.8% 6, 31.6%	2, 10.5% 3, 17.6% 11, 64.7%
Adjuvant Treatment Radiation Only Concurrent Chemoradiation No Adjuvant	18 9 9	13, 72.2% 3, 33.3% 3, 33.3%	5, 27.8% 6, 66.7% 6, 66.7%

* Significant group differences (*p* < 0.05).

**Table 3 curroncol-32-00562-t003:** Clinical assessment results—vitals, body composition, physical function (n = 19).

Exam Component	Result (Mean ± SD)
Resting Heart Rate, bpm	77.4 ± 8.7
Resting Blood Pressure, mm Hg Systolic Blood Pressure Diastolic Blood Pressure	123/79 123.0 ± 9.4 78.6 ± 7.1
Height, kg	174.6 ± 11.3
Weight, cm	80.8 ± 14.9
BMI, kg/m^2^	26.5 ± 3.9
**SPPB** Balance Score, out of 4 Gait Speed Score, out of 4 Gait Aids: walker (n = 4), cane (n = 5), none (n = 44) Chair Stand Test Score, out of 4 Total Score, out of 12	3.8 ± 0.7 3.9 ± 0.4 3.4 ± 1.0 11.0 ± 1.6

**Table 4 curroncol-32-00562-t004:** Clinical assessment results—shoulder and neck function (surgical vs. non-surgical side).

Category	Surgical Side (Mean ± SD)	Non-Surgical Side (Mean ± SD)
**Spinal Accessory Nerve (CN XI)-Innervated Muscles (MRC scale, /5 ± SD)**		
Sternocleidomastoid	5.0 ± 0	5.0 ± 0
Trapezius	5.0 ± 0.2	5.0 ± 0
**Upper Extremity Strength (myotome) (MRC scale, /5 ± SD)**		
Deltoids/Shoulder Abduction (C5)	4.9 ± 0.3	5.0 ± 0
Biceps/Elbow Flexion (C6)	5.0 ± 0	5.0 ± 0
Wrist Extension (C7)	5.0 ± 0	5.0 ± 0
Triceps/Elbow Extension (C8)	5.0 ± 0	5.0 ± 0
Intrinsics/Finger Abduction (T1)	5.0 ± 0	5.0 ± 0
Neck Extension	5.0 ± 0 affected vs. unaffected side N/A	
Neck Flexion	4.9 ± 0.2 affected vs. unaffected side N/A	
**Shoulder Active Range of Motion (mean degrees** ± SD)		
Abduction	154.2 ± 26.7	151.6 ± 32.1
Flexion	158.2 ± 12.3	160.1 ± 11.7
External Rotation	72.9 ± 18.1	80.8 ± 10.5
Internal Rotation	51.4 ± 19.6	48.1 ± 23.2
**Scapular Winging—number (%)**		
Yes	8, 32.0%	1, 7.7%
No	17, 68.0%	12, 92.3%
**Neck Active Range of Motion (mean degrees** ± SD)		
Flexion	43.8 ± 14.2 surgical vs. non-surgical side N/A	
Extension	45.7 ± 12.2 surgical vs. non-surgical side N/A	
Lateral Rotation	65.4 ± 23.8	57.4 ± 11.8
Lateral Flexion	33.6 ± 13.1	31.8 ± 11.4

Surgical Side = strength or sensory examination results on the same side as neck dissection. Non-Surgical Side = strength or sensory examination results on the opposite side of the neck dissection.

**Table 5 curroncol-32-00562-t005:** Functional outcomes comparing participants who attended vs. not attended.

	Baseline		6 Months		12 Months	
Questionnaire	Attended (n = 16)	Not Attended (n = 14)	Attended (n = 13)	Not Attended (n = 12)	Attended (n = 13)	Not Attended (n = 8)
Shoulder Function (QuickDASH)	17.4 ± 8.4	20.9 ± 9.0	14.9 ± 3.9 *	19.2 ± 4.5	14.0 ± 4.0	18.9 ± 10.8
Neck Function (NDII)	18.7 ± 10.8	17.7 ± 6.4	14.3 ± 5.8	19.1 ± 7.0	14.0 ± 5.3	17.1 ± 10.6
Swallow Function (EAT-10)	12.1 ± 11.0 **	12.1 ± 10.4	4.6 ± 6.0 **	12.8 ± 12.5	6.3 ± 10.2	8.8 ± 11.7

* denotes significant difference (*p* < 0.05) between groups. ** denotes significant difference (*p* < 0.05) within groups across timepoints. QuickDASH, scores/100. Higher scores denote higher levels of dysfunction; NDII, scores/100. Higher scores denote better shoulder function and less impairment; EAT-10, scores/40. Higher scores denote higher levels of swallowing dysfunction.

## Data Availability

The data presented in this study are available on request from the corresponding author.

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
