# Peer review of "Feasibility of a Physiatry Assessment Clinic to Address Physical Impairment in Head and Neck Cancer Patients Following Neck Resection and Free Flap Reconstruction"

_curroncol, 2025, doi:10.3390/curroncol32100562_

Round 1

Reviewer 1 Report

Comments and Suggestions for Authors

Yes, head and neck carcinoma is devastating and life-changing. Having therapeutic possibilities that can improve quality of life is very valuable.
That's why the work is interesting: it has an adequate introduction and clear objectives. Here's the first point for improvement: the objective is clear, but the conclusions aren't focused on answering the question; they deserve reformulation.
The methods are well described, but here's a question for improvement: how many patients were examined to select 39? This statement is needed.
The results are detailed, but Tables 4 and 5 aren't self-explanatory... it's not clear to read that they provide the data the authors want. They need to be revised to include this communication more directly.
The discussion is pertinently focused on what to do with patients with serious illness and elegantly address the limitations of the work.

I believe the work is adequate in form and content, but needs minor revisions (focus on the conclusions and more communicative tables) for publication.
It should be published.

Author Response

Comment #1

Yes, head and neck carcinoma is devastating and life-changing. Having therapeutic possibilities that can improve quality of life is very valuable. That's why the work is interesting: it has an adequate introduction and clear objectives. Here's the first point for improvement.

The objective is clear, but the conclusions aren't focused on answering the question; they deserve reformulation.

Thank you for this feedback. We understand this comment refers to the conclusion within our abstract. The primary outcome of our study was to examine the feasibility of a Physiatry Assessment Clinic (PAC). While we believe our conclusion partially addresses this objective, we have updated the abstract to improve clarity (lines 53-56): “While recruitment to the PAC and assessment completion demonstrated feasibility, attendance posed challenges for patients. These findings highlight the need for innovative approaches to screening patients and tailoring rehabilitation services based on physical impairment.”

Comment #2

The methods are well described, but here's a question for improvement: how many patients were examined to select 39? This statement is needed.

Thank you for this feedback. We have updated the methods section to better reflect our methods regarding recruitment, which was based on the timing of the PAC clinic. Please see lines 124-127.

“The PAC accepted referrals from February 2022-September 2023, and therefore all patients involved in the larger parent study during this period were approached to attend the PAC. Reasons for not attending were tracked and characteristics of both attending and non-attending groups were recorded to asses for feasibility of the PAC. Of those who attended, assessment completion was tracked to assess feasibility.”

This was also updated in the methods on lines 216-217: “During the PAC recruitment period, a total of 36 individuals in the parent study were eligible and approached.”

Comment #3

The results are detailed, but Tables 4 and 5 aren't self-explanatory... it's not clear to read that they provide the data the authors want. They need to be revised to include this communication more directly.

Thank you for this feedback. We have updated the methods section to better reflect the purpose of these assessments. Please see lines 140-145: “Feasibility of the PAC was the primary outcome, assessed by comparing characteristics between those who attended versus those who did not attend. Exploratory analyses were completed to describe physical function, fitness, and shoulder, neck, and swallowing function between those who did and did not attend the PAC to understand whether the PAC recruited the participants with impairment at a time of need and if it was feasible for those with higher needs to attend the PAC.”

We have also updated the results section to better describe the purpose of table 4. Please see lines 242-244: “Table 4 outlines shoulder and neck function. As described in the methods section, instead of describing left versus right shoulder values, results are described by surgical side to assess possible differences in function.  There were no differences (p>0.05) between the surgical and non-surgical sides when evaluating neck and shoulder strength, range of motion, and scapular winging.”

Table 5 is meant to identify differences between the attended and non attended groups regarding function. We felt the current explanation below table 5 describes the tools and provides clarification, but have updated the title of the table for further clarity.

Please see lines: 258.

Comment #4
The discussion is pertinently focused on what to do with patients with serious illness and elegantly address the limitations of the work. I believe the work is adequate in form and content, but needs minor revisions (focus on the conclusions and more communicative tables) for publication.
It should be published.

We thank reviewer #1 for their comments and feedback and hope the revisions made improve clarity.

Reviewer 2 Report

Comments and Suggestions for Authors

Dear Authors,

I am glad that I've seen such a topic in the literature due to the facial and lifestyle changes that are induced after oncological and surgical treatment in the head and neck area. However, I have some suggestions:

  • language needs improvements (ex., rows 23-25)
  • reports such as 8.6 (SD=3.6) are usualy noted 8.6 ±3.6 
  • Rows 32-34 need rephrasing to increase the clarity of the study's aim

Introduction: GLOBOCAN has a later edition (2022/2024). Also, the references need to be the most recent ones.

Material and Methods: What period of time did you use to include the patients? It is not stated. Why did you include only 36 patients?

Various histological types require different surgery aproach; for example, BCC does not need neck disection or bone cancer need more extensive surgery (table 2). Hence, the PAC is different for different cohorts.

what staging did you use?

Do all of your patients receive a free flap? It is also important to examine the donor site, but maybe in future studies. Also, did any of the free flaps require revision?

How long were the patients hospitalized?

It may be beneficial if you summarize in a table (or supplementary) the physical and psychological examination (ex., questions and measurements)

Author Response

Comment #1

Dear Authors,

I am glad that I've seen such a topic in the literature due to the facial and lifestyle changes that are induced after oncological and surgical treatment in the head and neck area. However, I have some suggestions:

Language needs improvements (ex., rows 23-25).

We thank the reviewer for this comment. We have made changes to our wording in rows 23-25 (tracked changes in the manuscript file) as well as copied the changes here:

A growing body of evidence supports the role of early rehabilitation in improving patient outcomes; however, the optimal timing and implementation of rehabilitation services have not been established.

Comment #2

Reports such as 8.6 (SD=3.6) are usually noted 8.6 ±3.6.

We have updated our reporting of standard deviations to include “±” across all results in the manuscript.

Comment #3

Rows 32-34 need rephrasing to increase the clarity of the study's aim.

We have edited these rows (now row 35-40) to clarify the study’s aim: The purpose of this study was to assess the feasibility of a Physiatry Assessment Clinic (PAC) following neck resection and free flap reconstruction, during which physical function was assessed.

Comment #4

Introduction: GLOBOCAN has a later edition (2022/2024). Also, the references need to be the most recent ones.

Thank you very much for this information. We have now updated the GLOBOCAN citation to the 2024 reference. To the best of our knowledge, all other citations are up to date.

Comment #5

Material and Methods: What period of time did you use to include the patients? It is not stated. Why did you include only 36 patients?

Thank you for this feedback. We have updated the methods section to better reflect our methods regarding recruitment, which was based on the timing of the PAC clinic. Please see lines 124-127.

“The PAC accepted referrals from February 2022-September 2023, and therefore all patients involved in the larger parent study during this period were approached to attend the PAC. Reasons for not attending were tracked and characteristics of both attending and non-attending groups were recorded to asses for feasibility of the PAC. Of those who attended, assessment completion was tracked to assess feasibility.”

This was also updated in the results on lines 215-217: “During the PAC recruitment period, a total of 36 individuals in the parent study were eligible and approached.”

Comment #6

Various histological types require different surgery approach; for example, BCC does not need neck dissection or bone cancer need more extensive surgery (table 2). Hence, the PAC is different for different cohorts. What staging did you use?

We appreciate this point and have clarified in the methods that we used the American Joint Committee on Cancer 8 (AJCC8) system for staging. We have clarified this in Table 2.

Comment #7

Do all of your patients receive a free flap? It is also important to examine the donor site, but maybe in future studies. Also, did any of the free flaps require revision?

To be included, all patients received a free flap. We did not examine donor site in the assessment clinic, but agree this would be an important future consideration and have included this in the conclusions. Please see lines 346-347 “Future research should also include additional variables such as length of hospital stay, detailed description of donor site recovery, which was not collected in this study.”

Comment #8

How long were the patients hospitalized?

We did not collect this data, but agree this would be an important future consideration and have included this in the conclusions. Please see lines 346-347 “Future research should also include additional variables such as length of hospital stay, detailed description of donor site recovery, which was not collected in this study.”

Comment #9

It may be beneficial if you summarize in a table (or supplementary) the physical and psychological examination (ex., questions and measurements)

Thank you for this comment. We originally had this information summarized, but based on previous feedback and discussion with co-authors, felt it did not contribute to the purpose of the paper and distracted from the aim of feasibility. We therefore feel it is best to not include this information as it did not align with the primary aim of feasibility.